# No Double Descent in Self-Supervised Learning

**Dulhan Jayalath, Alisia Lupidi, Yonatan Gideoni** *
Department of Computer Science and Technology
University of Cambridge
{dhj26,aml201,yg403}@cl.cam.ac.uk

## Abstract

Most investigations into double descent have focused on supervised models while the few works studying self-supervised settings find a surprising lack of the phenomenon. These results imply that *double descent may not exist in self-supervised models*. We show this empirically in two additional previously unstudied settings using a standard and linear autoencoder. We observe the test loss has either a classical U-shape or monotonically decreases, rather than exhibiting the double-descent curve. We hope that further work on this will help elucidate the theoretical underpinnings of this phenomenon.[1]

## 1 Introduction and Background

Double descent is a hallmark phenomenon of deep learning, showing the rift between classical learning theory and the generalization capabilities of deep learning systems (Belkin et al., 2019). It manifests as a model's test loss deviating from the classical U-shape in the bias-variance tradeoff as the model becomes overparameterized (Neal et al., 2018; Belkin et al., 2019; Nakkiran et al., 2019). This form of the phenomenon is known as model-wise double descent, where once the model's number of parameters is increased the test loss reaches a local maxima known as the interpolation peak. This peak occurs when the number of parameters is approximately equal to the product of the dataset's size and the dimension of the model's output, where the model can *just* interpolate the data (Belkin et al., 2019). After passing the peak the model improves, resulting in the training loss decreasing further and the test loss monotonically decreasing as model capacity increases.

However, previous works study almost only supervised settings. This raises the question, what happens in the self-supervised case? The few works that asked this found that double descent is not as universal as it seemed. For example, Gedon et al. (2023) showed analytically and empirically that double descent does not occur in PCA. Luzi et al. (2021) found that Generative Adversarial Networks (GANs) do not exhibit it either, except in an artificial pseudo-supervised setting. Dar et al. (2020) contrasted the unsupervised and supervised settings by showing that when going from the former to the latter in subspace fitting tasks, double descent gradually appears. Dubova (2022) observed double descent in an autoencoder (AE) but the implications of their results are not obvious as it is unclear whether their task is supervised and the location of their interpolation peak seems to be independent of the model's capacity.

Following these findings, we hypothesize that *double descent does not occur in self-supervised settings*. We provide evidence for this claim through an empirical analysis of an autoencoder (AE) (Kramer, 1991; Hinton et al., 2011), showing that it does not exhibit double descent. We choose an AE because of the tension with Dubova (2022)'s results. We hope that our findings will help elucidate the theory of overparameterized models.

---

*All authors contributed equally.

[1]Code can be found at https://github.com/YonatanGideoni/double_descent_tiny_paper.

## 2 EXPERIMENTS

We search for double descent when training an AE on artificial data, akin to the analysis done in Gedon et al. (2023). A detailed overview of the experiment can be found in Appendix B. The dataset $\{x_i\}_{i=1}^N$ is generated through the relation, $x_i = Dz_i + \epsilon_i$ where $z_i \sim \mathcal{N}(0, I_d)$ are the latent variables, $\epsilon_i \sim \mathcal{N}(0, I_n)$ are noise vectors, $n$ is the number of features, $d$ is the latent dimension of the data, and $D_{ij} \sim \mathcal{N}(0, r)$ is a random matrix used to project the latent features into a higher-dimensional space, where $r$ is a constant used to control the signal-to-noise ratio (SNR).

To learn the latent distribution of the data, we used an AE with fully-connected single-layer decoders and encoders, each with the same number of hidden units. The latent layer's width was varied independently of the hidden layers' widths because of uncertainty regarding where double descent would occur. Previous results (Dubova, 2022) imply that if double descent occurs in AEs then it might be independent of the hidden width. Although the interpolation peak is where the number of model parameters is equal to the dataset's size multiplied by the dimension of the output, it is unclear what is the output's size. For example, one could argue it is the AE's latent dimension or its number of features. This ambiguity is dealt with by using a wide range of hidden and latent dimensions in the AE to cover both cases. The results are shown in Figure 1. The test loss never exhibits double descent, with its specific behavior depending on how the model was overparameterized.

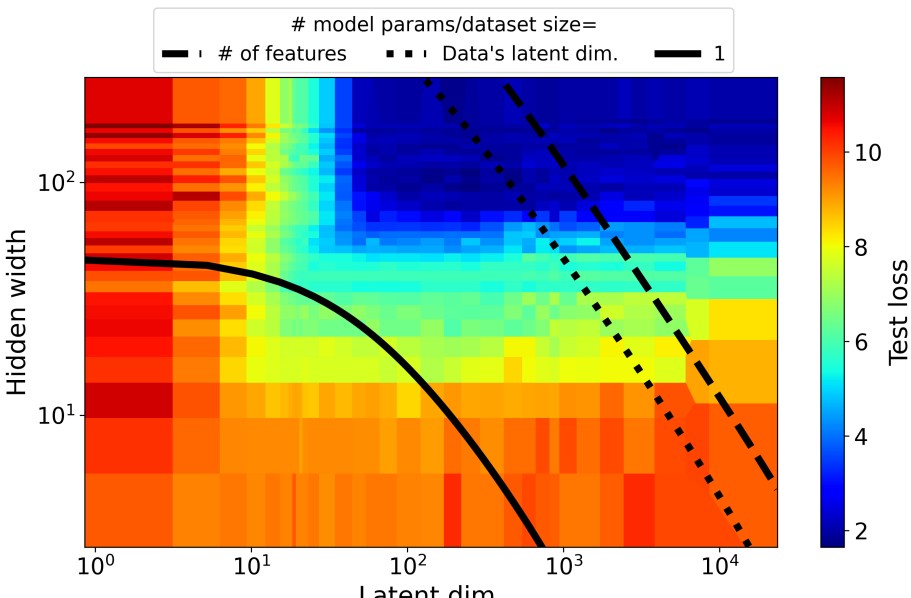

Figure 1: No double descent-like interpolation peak is observed across different latent and hidden widths. When going from a small latent dimension to a high one the loss has a bias-variance-like U-shape. However, when increasing the hidden dimension the loss only decreases. The black lines signify locations where the interpolation peak would be if we assume different output sizes.

We find that the test loss never exhibits double descent. Its specific behavior depends on how the model was overparameterized. An additional experiment with a linear AE is given in Appendix A.

## 3 DISCUSSION

We have found evidence that double descent does not occur in self-supervised settings. Our results add to the literature supporting this finding. The discrepancy between supervised and self-supervised settings may be due to their different formulations—while the former assumes noisy data, in the latter, the model tries to learn a distribution. A fundamental analysis and more extensive experiments are necessary to test this. As a theory is best understood through its shortcomings, we believe that building on these conjectures will help advance our understanding of overparameterized systems.

STATEMENT OF CONTRIBUTIONS

- **DHJ** ran and coded most of the experiments and contributed to writing.
- **AL** designed and ran experiments that weren't in the final paper and contributed to writing.
- **YG** designed the experiments, created the plots, and contributed to writing.

ACKNOWLEDGEMENTS

We thank Aditya Ravuri for a helpful discussion concerning VAEs and (in alphabetical order) Bobby, Yarin Gal, Ferenc Huszar, Lauro Langosco di Langosco, and Challenger Mishra for their insights about double descent and self-supervised models. We are grateful to Marina Dubova for insightful discussions about her paper. We also express our gratitude to Francisco Vargas and Chaitanya Joshi for reviewing our drafts.

URM STATEMENT

The authors acknowledge that at least one key author of this work meets the URM criteria of ICLR 2023 Tiny Papers Track.

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

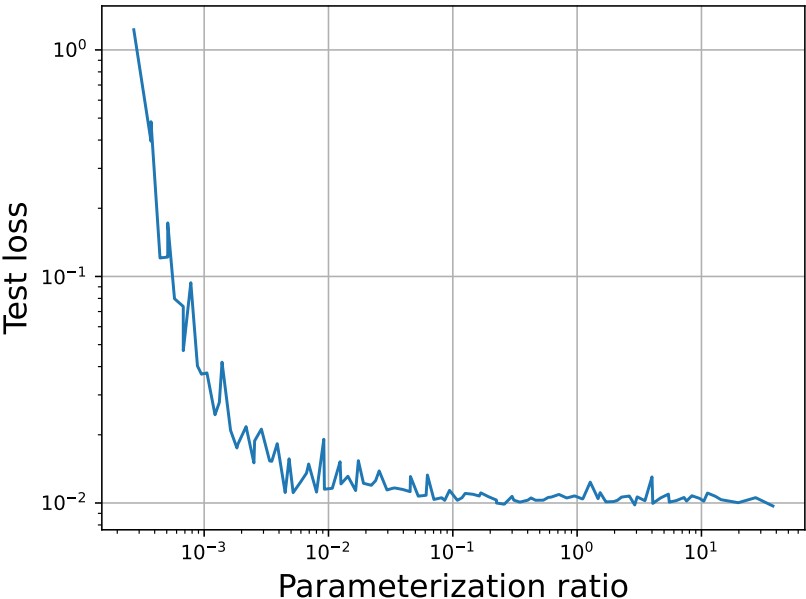

Figure 2: The test loss for different parameterization ratios in the linear AE experiment. Again, no double descent curve is seen.

## A    LINEAR AE EXPERIMENT

As double descent is also known to occur in linear regression (Nakkiran, 2019; Derezinski et al., 2019; Bach, 2023), it is natural to test our hypothesis on a similar task. This prompts the use of a linear autoencoder—an autoencoder with no non-linear activation functions. Given the ambiguity in the interpolation peak's location, this autoencoder's architecture was kept constant while the dataset size was varied, from being lower than the input dimension to far larger than the number of parameters. The resulting test loss, relative to different parameterization ratios, is shown in Figure 2. We find that there is still no discernible interpolation peak.

### A.1    EXPERIMENT DETAILS

The experiment's hyperparameters are given in Table 1. In this setting, the model had $29,445$ trainable parameters.

| Parameter Name | Value |
|---|---|
| Learning rate | 0.001 |
| Epochs | 1000 |
| Batch size | 20 |
| Data's latent dimension, $d$ | 10 |
| Number of features, $n$ | 25 |
| AE hidden dimension | 100 |
| AE latent dimension | 20 |
| Train dataset size, $N$ | $8 \cdot 10^0 - 1.1 \cdot 10^6$ |
| Test dataset size | 1000 |

Table 1: Parameters used for the linear AE experiments.

## B    EXPERIMENT DETAILS

### B.1    PARAMETER VALUES

The parameters used for the data generation and training procedures are given in Table 2. The optimizer used for training was Adam (Kingma & Ba, 2014) and the reconstruction loss was MSE.

| Parameter Name | Value |
|---|---|
| Learning rate | 0.001 |
| Epochs | 200 |
| Batch size | 10 |
| Data's latent dimension, $d$ | 20 |
| Number of features, $n$ | 50 |
| Train dataset size, $N$ | 5000 |
| Test dataset size | 10000 |
| SNR | 10 |

Table 2: Parameters used for the AE experiments.

### B.2    SNR

To control the SNR, $r$ is chosen such that $\frac{\mathbb{E}[||Dz||_2]}{\mathbb{E}[||\epsilon||_2]}$ has the desired value. As all the random variables are Gaussian, this amounts to setting $r = \frac{\text{SNR}^2}{d}$.

### B.3    PLOT DETAILS

The parameter space shown in Figure 1 was not sampled in a linear or logarithmic grid because we expected some regions to have a higher chance of exhibiting double descent than others. To show all the results at once, we colored the parameter space based on the loss of the closest measured point. The logarithmic distance is used instead of the normal one, so 100 is closer to 1000 than it is to 1.

### B.4    TRAINING LOSS

The training loss corresponding to the results shown in Figure 1 is given in Figure 3. The autoencoder transitions to the interpolation regime in a large part of the parameter space, which is necessary for DD to occur. This transition happens when the latent dimension is about 20, which is the latent dimension of the data, or when the hidden width is about 28. This could imply that interpolation is possible only when the autoencoder does not have an information bottleneck induced by its layers' widths.

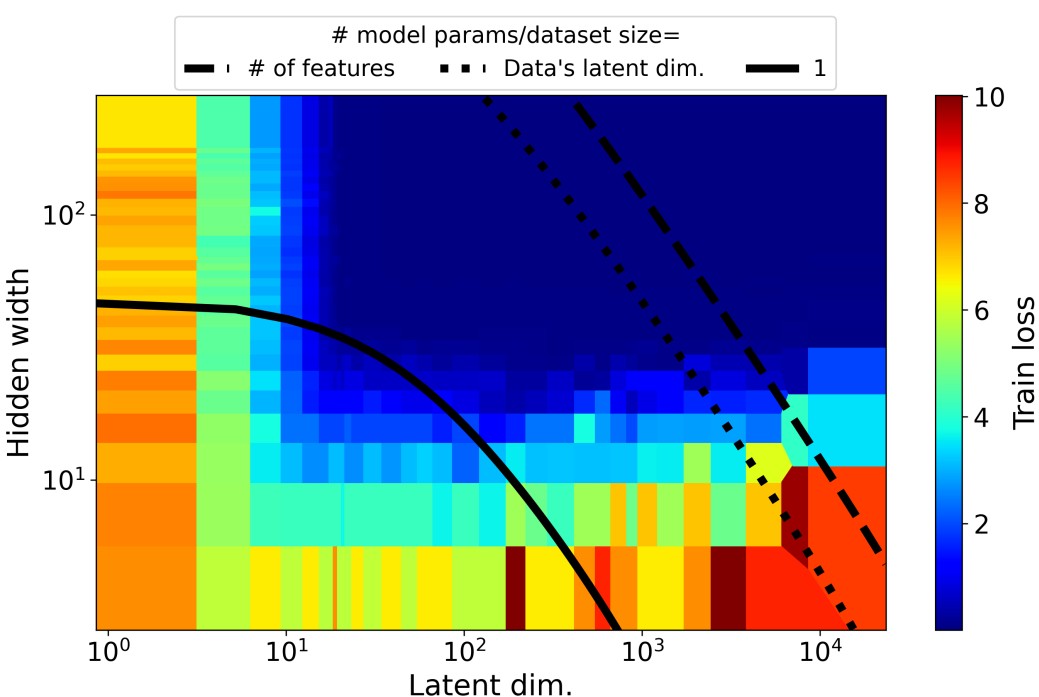

Figure 3: The train loss for the experiment discussed in section 2.

