# OpenReview forum: "No Double Descent in Self-Supervised Learning"
_ICLR.cc/2023/TinyPapers — Submitted to Tiny Papers @ ICLR 2023_

### Official Review · Reviewer_KDcC · 2023-03-27

**Confidence:** 4

**Summary Of Contributions:**

Double descent (DD) is an interesting phenomenon in machine learning (or, in specific, overparameterized models), where the test loss increases and decreases as the model size grows. However, it was mostly analyzed in the context of supervised learning. The paper argues that double descent doesn't happen in self-supervised learning with an experiment.

**Rating:**

Great Start (GS): a submission which meets some of the reviewing criteria but has room for improvement

**Strengths And Weaknesses:**

*Strengths*
- The findings are clearly communicated, and the paper appropriately discusses other relevant literature.
- The topic of the paper is relevant to the ICLR community.

*Weaknesses*
- Not much theoretical discussion on why DD might not occur in an unsupervised learning setup.
- The argument is based on a single experiment, even though it is not straightforward how to vary the complexity of the model.

**Suggested Changes:**

- I believe the authors are tackling an interesting problem, and the research is essential. I have some suggestions for the experiments. Double descent can be observed in an overparametrized linear regression model, where the exact solution can be computed. Could the authors experiment on a linear autoencoder setup? Regarding how to vary the model's complexity, the authors can fix the dimension and vary the number of training examples to see if DD happens; this allows to separate regions where N >> D, N ~= D, and N << D.
- I would be surprised if DD does not happen in the above setup since just optimizing the encoder makes the problem to be the same as linear regression (ignoring the interaction with decoders). However, including this will make the justification more convincing.

---

### Author Response · Authors · 2023-05-29
**Response to reviewers**

Thanks to the reviewers for their comments. We have made the following changes to address these points:

- As requested by reviewer KDcC, we have conducted an experiment with a linear autoencoder, varying the number of data samples while keeping the network's parameters the same. In this case also, we find no evidence of double descent.
- We have reworked our discussion in light of reviewer kJwe's comments.

---

### Comment · Area_Chair_kJwe · 2023-06-06
**Check for Archival**

This work meets the threshold for archival, contents the URM statement and is deanonymized.

---

### Meta-Review · Area_Chair_kJwe · 2023-04-02

**Recommendation:** Invite to archive
**Confidence:** 4

**Metareview:**

This paper studies the double descent (DD) phenomena in the self-supervised learning setting. The authors assume DD does not happen in the self-supervised case, and provide empirical justifications through tiny experiments with an autoencoder.

The AC agrees that the studied problem is crucial, and the motivation of this paper is clear. However, the theoretical insights behind the finding are unclear (Minor, we do not expect that in only 2 pages). The empirical justifications are not strong enough as pointed out by the reviewer (Major, please revise the paper carefully according to the reviewers' suggestions).

Overall, based on the review criteria of the ICLR TinyPaper Track, the clarity and correctness of this paper need to be improved. Despite the limitations, it is a great start for the authors to further dive into this crucial research problem.



**Summary:**

This paper studies the double descent (DD) phenomena in the self-supervised learning setting. The authors assume DD does not happen in the self-supervised case, and provide empirical justifications through tiny experiments with an autoencoder.

**Comments And Feedback To The Authors:**

> "While our experiment analyzes a relatively simple system, we believe our findings are valid because of the generality that double descent exhibits in similar supervised models" - Not convinced.

Please conduct the extra experiments suggested by the reviewer in the next version.

**Reason For Not Giving A Higher Recommendation:**

The clarity and correctness of this paper need to be improved. The empirical justifications are not strong.

**Reason For Not Giving A Lower Recommendation:**

The motivation is clear. The studied problem is crucial and relevant to the ICLR community.

---

### Decision · Program_Chairs · 2023-04-07

Invite to archive